# Neural mechanisms for the localization of unexpected external motion

Suma Chinta ®[1,2] & Scott R. Pluta ®[1,2] ✉

To localize objects during active sensing, animals must differentiate stimuli caused by volitional movement from real-world object motion. To determine a neural basis for this ability, we examined the mouse superior colliculus (SC), which contains multiple egocentric maps of sensorimotor space. By placing mice in a whisker-guided virtual reality, we discovered a rapidly adapting tactile response that transiently emerged during externally generated gains in whisker contact. Responses to self-generated touch that matched self-generated history were significantly attenuated, revealing that transient response magnitude is controlled by sensorimotor predictions. The magnitude of the transient response gradually decreased with repetitions in external motion, revealing a slow habituation based on external history. The direction of external motion was accurately encoded in the firing rates of transiently responsive neurons. These data reveal that whisker-specific adaptation and sensorimotor predictions in SC neurons enhance the localization of unexpected, externally generated changes in tactile space.

Much of our interaction with the world occurs through the volitional movement of our sensory organs. For example, primates move their eyes to visualize a scene, and rodents move their whiskers to explore nearby objects. To accurately localize objects, animals must differentiate the expected consequences of their actions, induced by active sensing, from real-world object motion. What neural mechanisms underlie this ability?

To address this question, we focused on the midbrain superior colliculus (SC), which contains multiple egocentric maps of sensorimotor space[1–4]. Many SC-mediated behaviors involve spatial processing, such as pursuing prey, escaping predators, or simply orienting toward the appropriate object[5–11]. While visually tracking objects, the SC helps stabilize the visual field and adjust for discrepancies between the predicted and real-world outcome of eye movements[12,13]. SC neurons are also highly selective to externally generated visual motion[14–16]. Given these behavioral and neurological insights, we hypothesized that SC neurons are specialized for localizing external motion during active touch. Such computations are likely critical for pursuing moving objects such as prey[17–20].

While decades of research have outlined SC function during visuospatial processing, remarkably less is known for somatosensation,

particularly during active touch[21–24]. The mouse SC is known to respond to whisker stimuli, yet nearly all published work in this area has been performed under anesthesia[4,25–28]. These experiments reveal that the SC contains a somatotopic map of whisker space, whereby individual neurons possess large (multi-whisker) receptive fields that are also responsive to artificial whisking. Therefore, active whisking in the mouse SC provides a tractable and underutilized model for revealing the cellular and circuit mechanisms that support innate tactile-guided behaviors.

Sensory responses in the SC have primarily been characterized during sensory fixation, with evidence showing that changes in egocentric heading (eye position) modulate response magnitude[21,22,24,29–31]. This approach reveals that sensory responses in the SC are modulated by extrasensory inputs. Additional work has shown that eye movements in darkness suppress SC activity, arguing for the presence of a motor prediction that functions to suppress self-generated visual motion[32,33]. Similarly, studies in humans reveal a perceptual attenuation of self-generated touch relative to an equivalent touch generated by external motion[34]. Tactile attenuation is thought to result from sensorimotor predictions that subtract the expected consequences of one's self-motion[34]. The somatosensory whisker system offers a

[1]Department of Biological Sciences, Purdue University, West Lafayette, IN, USA. [2]Purdue Institute for Integrative Neuroscience, Purdue University, West Lafayette, IN, USA. ✉e-mail: spluta@purdue.edu

powerful model for investigating how the brain differentially integrates external and self-generated stimuli since mice instinctively move their whiskers to palpate objects, and the precise dynamics of touch can be accurately measured using high-speed imaging and markerless tracking[35,36].

To reveal the neural mechanisms for localizing external motion during active sensing, we designed a tactile virtual reality that simulates whisker-guided navigation. While running on a treadmill, mice rhythmically touched a cylindrical surface that rotated at the same velocity as their locomotion. This created a tactile flow-field that simulates running along a wall, similar to optic flow, and generates tactile stimulation directly controlled by self-motion[37–39]. Periodically, the center axis of the flow-field translated horizontally along the mouse's whisking field, causing external world movement. A transient neural response selectively emerged when external motion gained contact with a whisker that did not match the animal's self-generated stimulus history. Self-generated gains in whisker contact that matched self-generated history evoked an attenuated response. The size of the transient response gradually decreased with repetitions in external motion, and the direction of external motion was accurately encoded in population-level firing rates.

## Results

### Simulating whisker-guided virtual reality

We designed a closed-loop tactile system that simulates whisker-guided exploration in mice. During an experiment, a head-fixed running mouse voluntarily whisked against a nearby cylindrical surface that rotated at the same velocity as its locomotion (Fig. 1a, left). A high-speed (500 fps) infrared camera and digital encoders recorded whisker kinematics, surface movements, and running speed while an opaque object and white noise obscured visual and auditory cues. After the mouse locomoted and rhythmically touched the surface for a distance (200 cm), the surface either translated rostrally, caudally or remained at the starting location with equal probability (Supplementary Movies 1 and 3). After staying at the rostral/caudal location for an equivalent locomotor distance, the surface returned to the center location (if applicable, Supplementary Movies 2 and 4). While running and touching the surface, mice innately adapted the position of their whiskers to track changes in surface location and maintained a stable whisking frequency across conditions (Fig. 1a, right, Supplementary Fig. 1). Since translation timing was determined by locomotor distance, variation in locomotion speed made translation onset unpredictable (Fig. 1b). As the surface moved into a new location, the firing rate of most neurons transiently increased, as shown in two example units (Fig. 1c). A 3-shank, 128-channel silicon probe recorded neural activity simultaneously across the intermediate and deep layers of the lateral SC, approximately 300–1000 microns below its surface (Fig. 1d, 12 mice, 873 neurons). About two-thirds of all recorded neurons displayed a significant response to surface movement (67 ± 6%, 12 mice, 578 neurons, α < 0.05, 1-way ANOVA).

### The transient response emerges during externally generated gains in whisker contact

To reveal the stimulus variables driving the transient response in SC neurons, we examined the dynamics of whisker touch during translations in surface location. We performed a combination of computerized whisker kinematic analysis and manual inspection of high-speed video. We discovered that surface movements elicited transient increases in firing rate only when they entered a location that gained (GoW) or lost (LoW) contact with a whisker (Fig. 2a, b, Supplementary Fig. 2a). The addition or subtraction of a single whisker from the surface was sufficient for this effect. Most neurons preferentially responded to gains in whisker contact (Supplementary Fig. 2b, GoW, 55 ± 4%, 8 mice, 385 neurons). Surface movement that pushed against the whiskers without gaining contact with a new whisker did not elicit a

transient response (Fig. 2a). In some neurons, the transient response was followed by persistent activity evoked by active whisking against the surface (Fig. 2b). Overall, the transient response was significantly larger than this self-generated activity (p < 0.0001, paired t-test, 529 neurons, 10 mice, Fig. 2c). To confirm that these responses were whisker-mediated, we repeated the experiment after trimming off the whiskers and found that nearly all neurons became unresponsive to external motion (59 ± 4% pre-trimming vs. 6 ± 1% post-trimming, 156 neurons, 2 mice, 1-way ANOVA, α < 0.05, Supplementary Fig. 2c). To understand the receptive field of SC neurons, we quantified each neuron's response to externally generated GoW and LoW stimulation (10 mice, 529 neurons, Fig. 2d). Almost half of all neurons displayed a transient increase in firing rate during both GoW and LoW (251/529 neurons) stimulation, while the remaining neurons increased their firing rate for one stimulus type yet decreased it for the other. To illustrate these effects, we plotted the population-averaged firing rates of each response type relative to the onset of external motion (Fig. 2e, Supplementary Fig. 2d–f). The GoW and LoW stimuli generated during surface movement likely caused changes in follicle tension that are comparable to the passive deflection and release of the whisker, which generate ON and OFF responses in the trigeminal ganglion, thalamus, and cortex[40,41].

To verify that transient response adaptation was whisker-specific, we selected trials where consecutive surface movements gained contact with different whiskers. With each additional gain in whisker contact, the transient response re-emerged from an adapted state (Fig. 2f, g). The magnitude of the transient response was not affected by the velocity of external motion (Supplementary Fig. 3), nor the absolute number of whiskers that gained surface contact (Supplementary Fig. 4). Overall, these data reveal a transient tactile response that selectively emerges for externally generated gains in whisker contact (external-GoW). Are SC neurons equally sensitive to self-generated gains in whisker contact (self-GoW)?

### SC neurons prefer external over self-generated gains in whisker contact

Since external-GoW stimulation in our experiment did not match the sensorimotor predictions built by the self-generated tactile history, we hypothesized that SC neurons prefer external- over self-GoW stimulation. To test his hypothesis, we modified our experiment to control the strength and rate of GoW stimulation. With this modified approach, the surface periodically moved between rostral whisker space and a location entirely outside the whisking field. With the surface in rostral whisker space, mice periodically touched (self-GoW) the surface as they voluntarily transitioned between quiescence (no movement) and active touch (Fig. 3a). By carefully analyzing high-speed video, we calculated neuronal firing rates relative to the onset of self-GoW stimulation, and also relative to gains in whisker contact generated during surface movement into rostral space (external-GoW). We carefully selected external-GoW trials where whisker touch was initiated during the protraction phase, removing any trials where the surface pushed against the already protracted whisker. Prior to each external-GoW stimulus, mice were freely whisking in air, building a stimulus history unique from active touch. However, between each self-GoW stimulus, mice were quiescent and therefore repeating the same stimulus history with each bout of active touch (Fig. 3a).

Almost all SC neurons preferred external- over self-GoW stimulation (Fig. 3b). Some neurons only responded to external-GoW stimulation, yet most neurons responded to both external- and self-GoW stimuli (Fig. 3b, c). Overall, population firing rates were significantly greater during external-GoW stimulation (p = 6e⁻¹³, Wilcoxon signed rank test, 4 mice, 139 neurons, Fig. 3c, e, Supplementary Fig. 5). Whisker curvature between both conditions was nearly identical, indicating that differences in stimulus strength did not cause this effect (Fig. 3d, Supplementary Fig. 5a). The angle of the whisker at

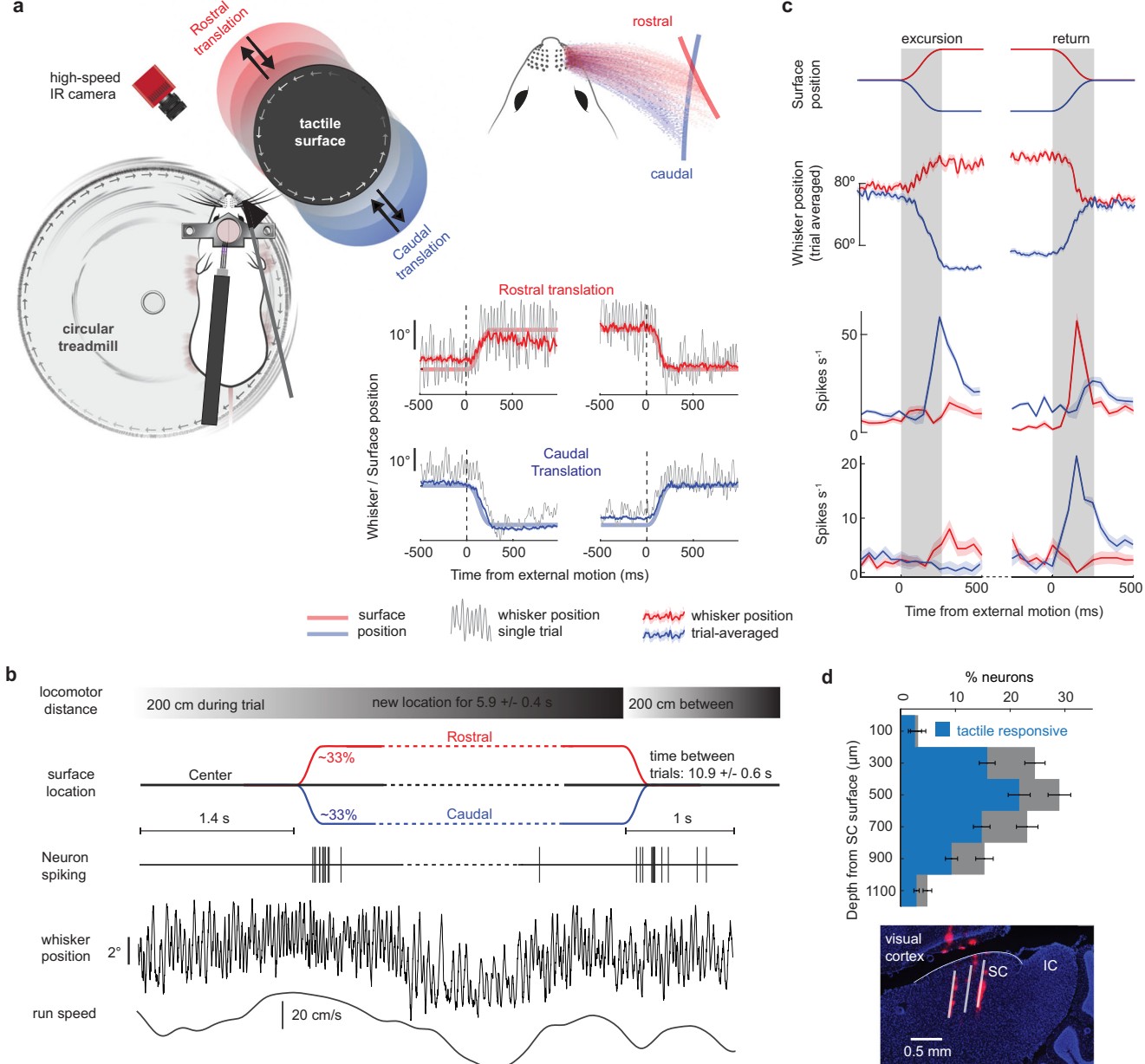

**Fig. 1 | Tactile and neural dynamics during whisker-guided virtual reality.**
**a** Schematic of whisker-guided virtual navigation, illustrating a head-fixed mouse locomoting on a treadmill and whisking against a surface. After trial initiation, there was an equal probability for the external tactile surface to translate into either rostral (in red) or caudal (in blue) whisker space or stay at the center position (in black). Upper right: single-whisker location density while the surface was at the rostral or caudal location. Bottom right: whisker position (mean ± s.e.m) and surface location (transparent line) during external surface movements in the rostral (top in red) and caudal (bottom in blue) direction. The first half is a surface excursion, while the second half is the return to the center position. The gray line is the whisker position in one example trial. **b** Trial structure aligning surface position, neural spiking, whisker position, and locomotion speed. **c** Trial-averaged whisker position and firing rate of two example SC neurons during each type of surface translation (50 ms bins, error bars represent s.e.m). The vertical gray bar indicates the period of surface movement. **d** Top: Percentage of tactile responsive SC neurons (blue) as a function of depth from the SC surface (12 mice, 873 neurons, error bars represent s.e.m). Bottom: dye labeling of the electrode shanks at the recording site in the intermediate and deep layers of the lateral SC. Source data are provided as a Source Data file.

touch onset had no influence on the external-GoW response (Supplementary Fig. 5c). To rule out adaptation as a potential mechanism for the larger external-GoW response, we calculated the time that preceded each GoW stimulus. Overall, the external-GoW stimulus occurred at a marginally faster rate (Mann–Whitney, $p = 0.04$, Fig. 3f, Supplementary Fig. 5b). Therefore, the rate of stimulation might predict a smaller external-GoW response, yet we observed the opposite.

## Self-generated stimulus history controls transient response magnitude

Although stimulus strength, whisker angle, and repetition rate could not explain the larger external-GoW response, locomotion speed was not rigorously controlled (Fig. 3g). In the external-GoW condition, the animal was already locomoting, yet in the self-GoW condition, the animal was transitioning into locomotion. While the effect of

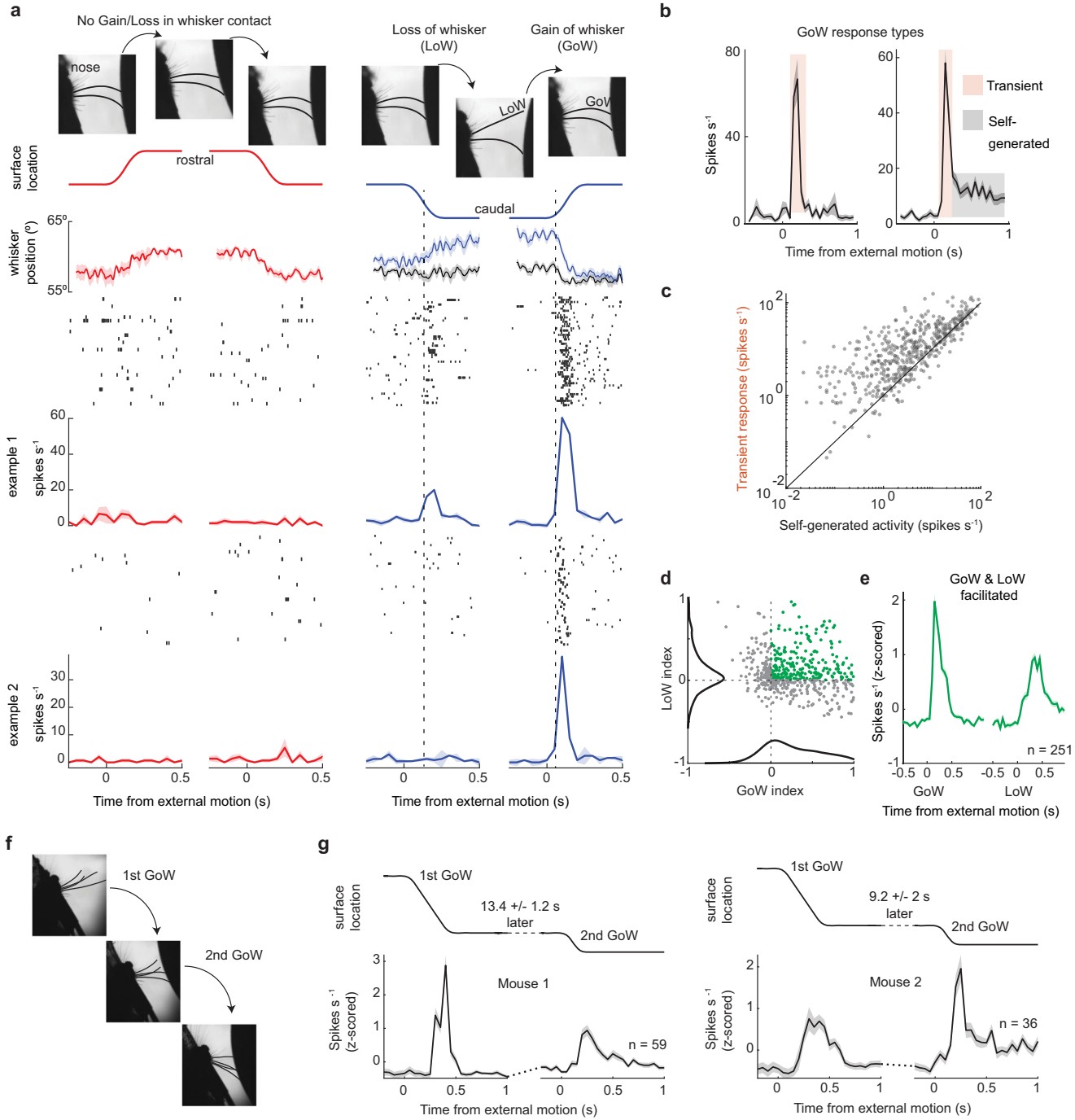

**Fig. 2 | The transient response emerges during externally generated gains in whisker contact. a** Surface location, whisker position and the firing rates of two example neurons during external surface movement into rostral or caudal space. The vertical dashed line marks the estimated moment of a gain/loss in whisker contact. **b** Two example neurons illustrating the two response types in SC neurons: a transient response evoked during external motion and sustained activity from self-generated touch. **c** Scatter plot comparing the transient and self-generated firing rates of SC neurons (10.4 ± 0.7 spikes/s for transient, 17.7 ± 1 spikes/s for self-generated activity; p = 526, two-sided paired-sample t-test; 10 mice, 529 neurons). **d** Scatter plot comparing the GoW and LoW modulation of each neuron (10 mice, 529 neurons). **e** Population-averaged firing rates of 251 neurons whose neural activity was facilitated for GoW and LoW stimulation. **f** High-speed imaging frames showing two consecutive GoW stimuli. **g** Population average firing rate of two mice during two consecutive GoW stimuli. All firing rates are binned at 50 ms; all values are mean ± sem. Source data are provided as a Source Data file.

locomotion on tactile responses in the SC is unknown, sensory responses are known to be weaker during whisking than during rest[42]. Therefore, greater locomotion and whisking potentially reduced the transient response. To eliminate any potential effect of locomotion, we devised a novel approach to rigorously control for locomotion and whisking. This approach also eliminated any potential differences in touch quality that may have gone undetected in Fig. 3 (Fig. 4).

In this experiment, mice only had one whisker contacting the surface. A trial was initiated with the mouse running and freely whisking in air. When the mouse reached a threshold distance on the treadmill and then stopped free-whisking, an object quickly (150 ms) entered its whisking field (Fig. 4a, see boxplot). Moments later, when the animal resumed whisking, it touched the extended object and generated an external-GoW event. The object then stayed in the

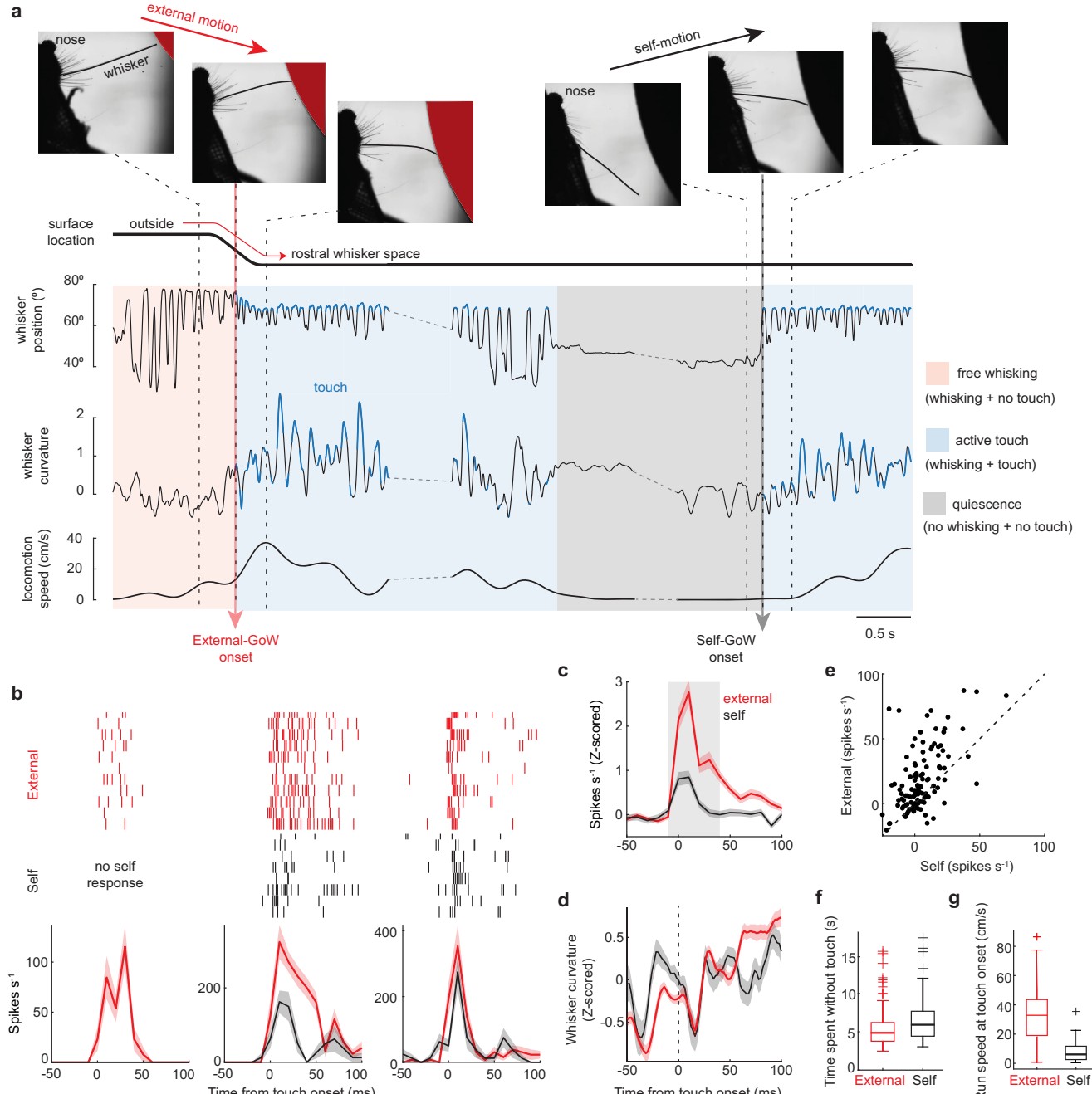

**Fig. 3 | SC neurons prefer externally generated gains in whisker contact.**
**a** Example trial illustrating the experimental design. Initially, the mouse freely whisked in air. After the mouse ran a set distance, external motion in rostral space generated whisker contact (external-GoW). After a period of whisking against the surface, the mouse voluntarily stopped and then restarted whisking to create self-generated touch (Self-GoW). Surface location, whisker position, whisker curvature, and locomotion speed of the animal during the example trial are presented. Vertical dashed lines represent the moment the mouse snapshots on the top are taken.
**b** Three neurons from one mouse comparing responses to external and self-GoW stimulation. Rasters and histograms of spiking aligned to touch onset. **c** Population-averaged, z-scored firing rates aligned to the onset of external- and self-GoW sti-mulation (4 mice, 139 neurons). **d** Population-averaged, z-scored whisker curvature aligned to the onset of external- and self-GoW. **e** Scatter plot comparing the change

in neuronal firing rates for external- and self-GoW stimulation (4 mice, 139 neurons, $p = 6e^{-13}$, two-sided Wilcoxon signed rank test). Firing rates are subtracted from their corresponding baselines. **f** Boxplots comparing the time mice spent without touch before an external- or self-GoW stimulus occurred ($p = 0.0351$, two-sided Mann–Whitney U-test, $n = 108$ external touches and 38 self-touches from 4 mice). **g** Boxplots comparing run speed at external and self-touch onset occurred ($p = 4.6e^{-7}$, two-sided Mann–Whitney U-test, $n = 108$ external touches and 38 self-touches from 4 mice). The central mark indicates the median. The bottom and top edges of the box indicate the 25th and 75th percentiles, respectively. The bottom and the top edges of the whiskers are the minima and maxima, excluding outliers. '+' indicates outliers. All values are mean ± s.e.m. All firing rates are binned at 10 ms. Source data are provided as a Source Data file.

whisking field for an extended period (47.1 ± 3.2 s), allowing the mouse to accumulate a stimulus history by periodically stopping and restarting active touch. Every time it restarted active touch, the first touch in each whisking bout generated a self-GoW event. After it reached the set treadmill distance, the object retracted, and the trial restarted with the mouse free-whisking in air. On average, the external-GoW response, which followed free-whisking, was significantly greater than the self-GoW response that followed a bout of active touch

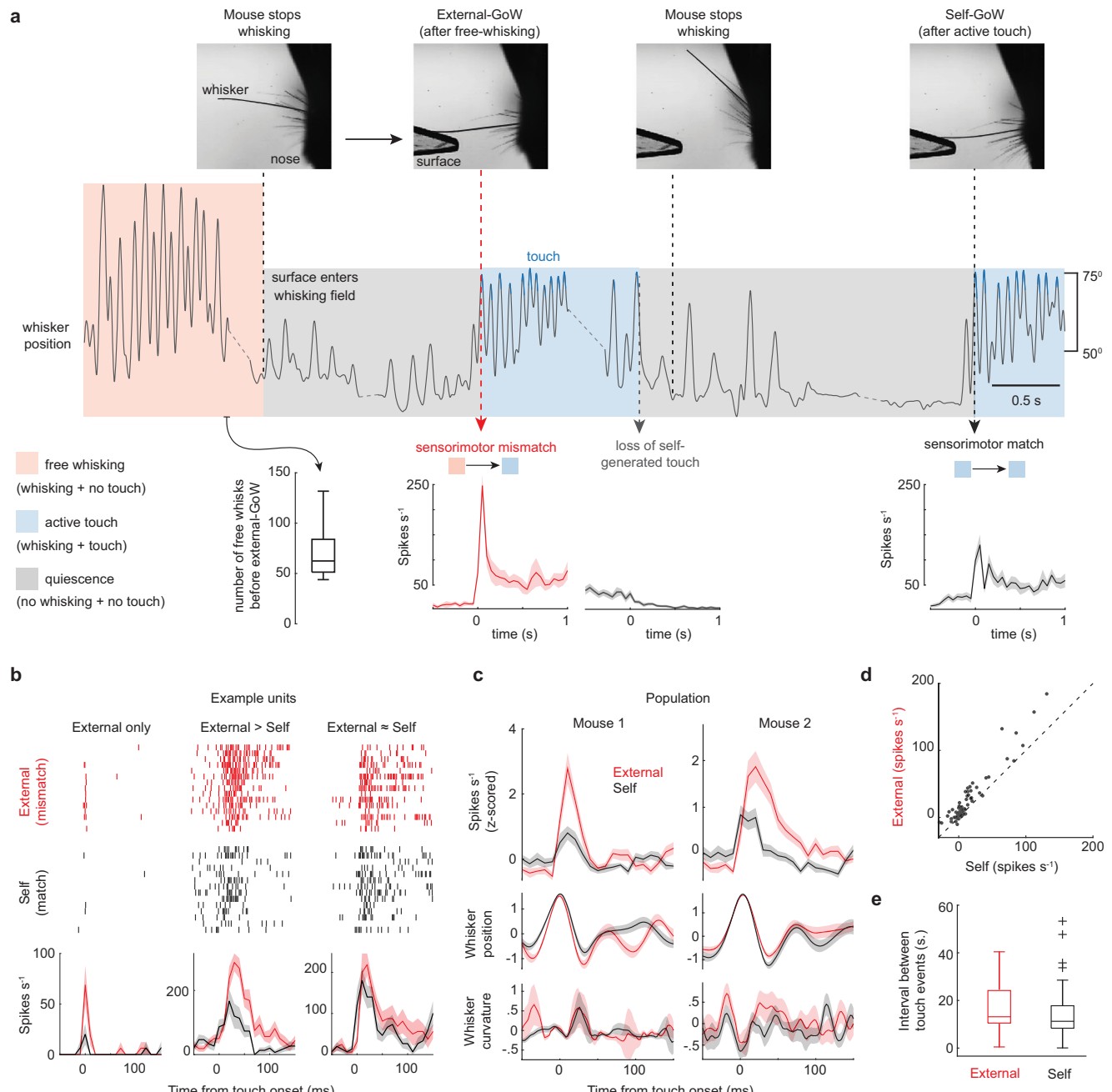

**Fig. 4 | Self-generated stimulus history controls transient response magnitude.**
**a** Schematic illustrating the experimental design. At the start of the trial, the mouse
was running and whisking freely in air (see boxplot for quantification). After the
mouse crossed a threshold locomotion distance and then stopped running, an
object entered its whisking field. The mouse voluntarily resumed whisking and
touched the object. The mouse then repeatedly stopped and restarted active touch.
After running and touching the object for a set distance, the surface retracted, and
the trial restarted with free-whisking. Bottom, histogram of firing rates for one
neuron during external- and self-GoW stimulation. **b** Rasters and histograms of
spike timing for three neurons from one mouse during external- and self-GoW
stimulation. **c** Population-averaged neuronal firing rates and whisking variables
aligned to the onset of touch (2 mice, 74 neurons). **d** Scatter plot comparing the
change in neuronal firing rates during external and self-GoW stimulation (2 mice, 74
neurons, $p = 2e^{-11}$, two-sided Wilcoxon signed rank test). **e** Boxplot comparing the
time mice spent without touch before external and self-GoW stimulation (two-
sided Mann–Whitney U-test, $p = 0.17$, 28 external-GoWs and 40 self-GoWs from 2
mice). The central mark indicates the median. The bottom and top edges of the box
indicate the 25th and 75th percentiles, respectively. The bottom and the top edges
of the whiskers are the minima and maxima, excluding outliers. '+' indicates out-
liers. All values are mean ± s.e.m. All firing rates are binned at 10 ms. Source data are
provided as a Source Data file.

(Fig. 4b–d, $p = 2e^{-11}$, $n = 74$, two-sided Wilcoxon signed rank test).
Therefore, the sensorimotor mismatch between the free-whisking sti-
mulus history and the external-GoW stimulus evoked a significantly
larger tactile response. Touch kinematics (whisker position and cur-
vature) were identical between the external- and self-GoW conditions

(Fig. 4c). Neurons with a larger self-GoW response were more strongly
modulated by external-GoW stimulation (Fig. 4d). Importantly, repe-
tition rate was equivalent between external- and self-GoW stimulation,
indicating that sensory adaptation could not explain the larger
external-GoW response (Fig. 4e, $p = 0.17$, Mann–Whitney).

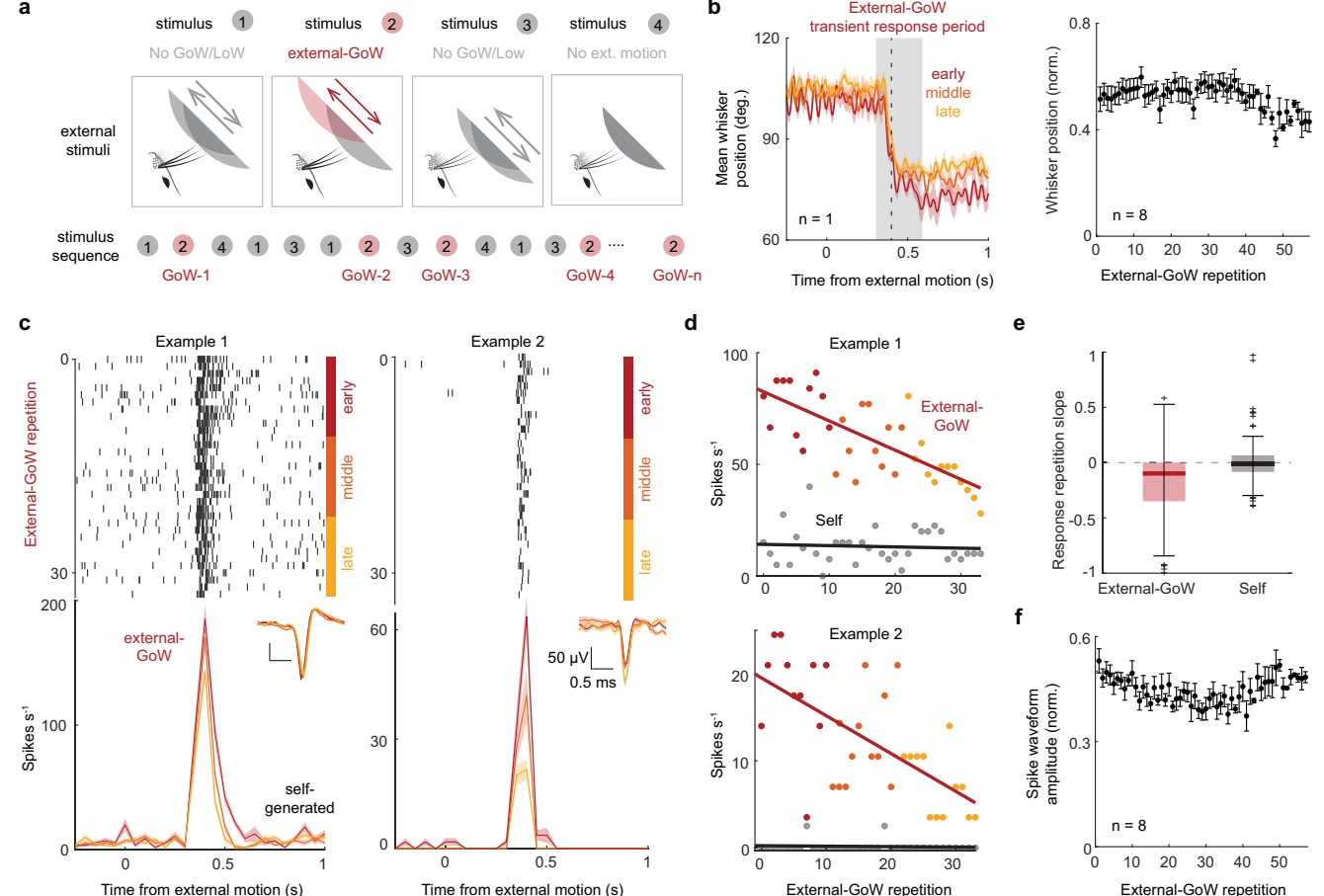

**Fig. 5 | Repetitions in external motion reduce transient response magnitude.**
**a** Schematic illustrating the different stimulus types and how they are randomly repeated over the course of the experiment. The return motion of the 2nd stimulus, which is a GoW, is selected for analysis in (**c**). **b** Left, trial-averaged whisker position relative to surface movement for early (first 1/3rd), middle, and late (last 1/3rd) trials of GoW stimulation (one mouse). Right, trial-averaged whisker position calculated during the GoW response period and plotted as a function of stimulus repetition (8 mice). Error bars represent s.e.m. **c** Rasters and histograms of spiking in two example neurons divided by early, middle, and late periods. Inset, spike waveforms during the same periods (color-coded). **d** Magnitude of the external-GoW and self-generated responses as a function of stimulus repetition in the same

two example neurons in (**c**). external and self-firing rates were obtained from transient and sustained windows described in "Methods". **e** Population slopes of the linear regression of external-GoW response (8 mice, 223 neurons, $p = 2e^{-12}$, one-sided $t$-test) and self responses (8 mice, 98 neurons, $p = 0.64$, one-sided $t$-test) as a function of stimulus repetition. The central mark indicates the median. The bottom and top edges of the box indicate the 25th and 75th percentiles, respectively. The bottom and the top edges of the whiskers are the minima and maxima, excluding outliers. **f** Normalized spike waveform amplitude as a function of stimulus repetition (8 mice) All values are mean ± s.e.m. Source data are provided as a Source Data file.

**External motion history controls transient response magnitude**
Next, we sought to determine if the magnitude of the transient response, evoked by external-GoW stimulation, was controlled by the history of external motion. To do so, we analyzed the effect of stimulus repetition on neuronal firing rates. For each experiment, we selected a surface movement that caused external-GoW stimulation (Fig. 5a). To ensure that any effects we observed were not caused by changes in whisking behavior, we analyzed the position of the whisker as a function of stimulus repetition. Across the population (8 mice), there was no notable change in whisking (Fig. 5b). However, we discovered that the magnitude of the transient response linearly decayed with increasing external-GoW repetitions, while activity related to self-generated touch was stable, as shown in two example neurons (Fig. 5c, d). The linear rate of transient response habituation was consistent across the population (Fig. 5e, 223 transient neurons, 8 mice, $p = 2e^{-12}$; 98 self-responsive neurons, $p = 0.64$, one-sided $t$-test). Spike waveform amplitude was stable in our recordings, indicating that a progressive decrease in spike detection cannot explain these effects (Fig. 5f, 5c insets). Habituation was unlikely caused by low-level sensory adaptation since the average interval between each external-GoW stimulus

was around 1 min (67 ± 5 s, 8 mice). Therefore, SC neurons appear to steadily habituate to repeated tactile stimulation occurring over the course of many minutes, providing evidence that external stimulus history controls transient response magnitude (See also Supplementary Fig. 6).

**The transient response encodes the direction of external motion**
To assess the function of neuronal selectivity for external motion, we tested the object localization accuracy of population-level SC activity over time. Using population decoding (see "Methods"), we discovered that SC activity accurately classified surface location (center, rostral, or caudal), with the highest accuracy during external motion, as shown in one example mouse (Fig. 6a, 99% accuracy, 76 neurons). Before the start of external motion, classification accuracy was at chance, indicating that neural activity could not predict the upcoming surface movement. In neurons that were activated by self-generated touch, classification accuracy was persistently high, revealing a stable representation of object location in this population (Fig. 6b, 30 neurons, 1 mouse). In neurons selective to external motion (transient responsive), classifier accuracy was equally high, but only during surface

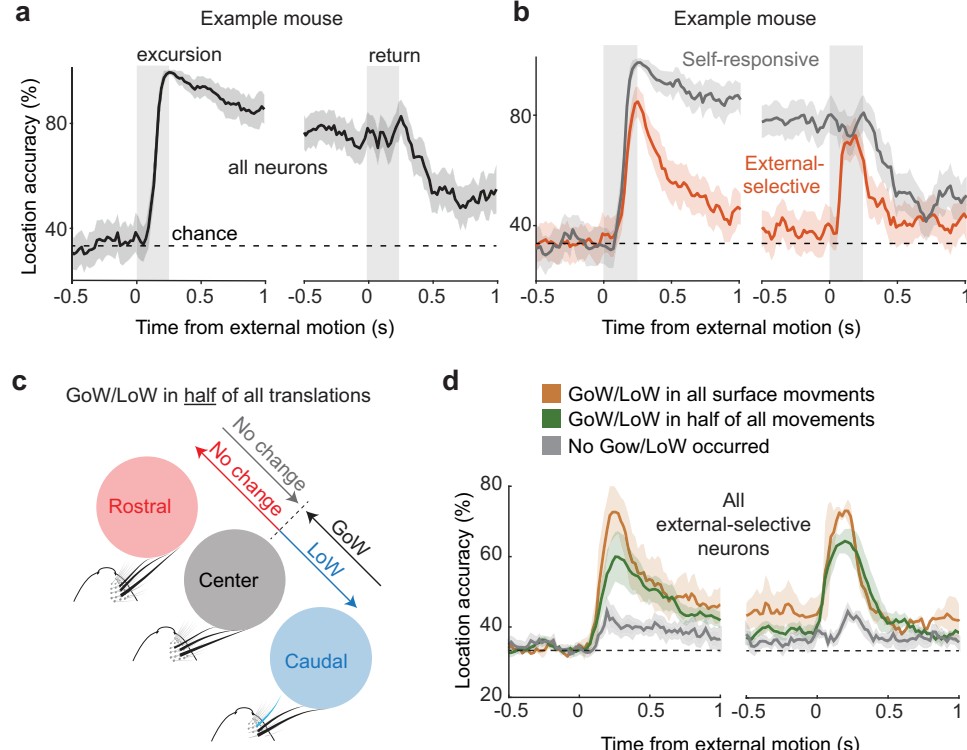

**Fig. 6 | The transient response encodes the direction of external motion.**
**a** Location decoding accuracy using all the tactile responsive neurons recorded in a single mouse (75 neurons). **b** Decoding accuracy in the same mouse with self-responsive (sustained, $n = 29$) and external selective (transient, $n = 34$) neurons separated. **c** Diagram illustrating an experiment where half of all the surface

translations caused a gain/loss in whisker contact. **d** Decoding accuracy for external selective neurons averaged across animals with the same number of gains/losses in whisker contact (GoW/LoW in all movements: 2 mice, 59 neurons; GoW/LoW in half of all movements: 6 mice, 96 neurons; Zero GoW/LoW movements: 4 mice, 33 neurons). All values are mean ± s.e.m. Source data are provided as a Source Data file.

movement, due to their rapid adaptation to self-generated touch (46 neurons, 1 mouse). We found that classifier accuracy was highest between surface locations that gained/lost whisker contact (Fig. 6c, d, Supplementary Fig. 7, GoW/LoW in all movements: 94 ± 5% accuracy, 2 mice; GoW/LoW in half: 71 ± 7%, 6 mice; GoW/LoW in no movements: 56 ± 3%, 4 mice). Transient responsive neurons had almost no information about external movements that did not gain or lose whisker contact. Therefore, a topographic representation of whisker space is likely important for encoding the location of external motion. We found evidence for such a map in one of our recordings where the rostral and caudal surface movements gained contact with different whiskers and preferentially activated their corresponding brain areas (Supplementary Fig. 8).

## Discussion

Active sensing requires the brain to contextualize incoming sensory stimuli with outgoing motor commands. To localize objects during active sensing, animals must recognize if changes in egocentric space originate from externally or internally generated movement[34,43]. By simulating whisker-guided exploration, we discovered neurons in the mouse SC that are specialized for locating unexpected, externally generated changes in tactile space. We identified a rapidly adapting response, which strongly preferred externally generated touch and that gradually habituated with repetitions in external motion. Rapid adaptation has been observed in the SC during passive visual stimulation[44–46] and passive, rhythmic deflection of the whiskers in anesthetized animals[28]. During our experiment, mice were performing foveal whisking at the onset of external-GoW stimulation. Therefore, the somatosensory state was possibly akin to visual foveation within a narrow field of search. We observed a diversity of adaptation rates across the population of recorded neurons, with many neurons only

transiently responding to externally generated touch and other neurons sustaining an equally strong response to self-generated touch. Importantly, we found that self-generated stimulus history was a primary factor controlling transient response magnitude. Therefore, the transient response reflects a sensorimotor mismatch between self- and external motion.

While the circuit mechanisms supporting rapid tactile adaption in the SC are unknown[47], both the barrel cortex and brainstem provide monosynaptic sensory drive to SC neurons[25–27,48,49]. Neurons in layer 5 of the barrel cortex show rapid sensory adaptation during externally generated surface whisking, similar to our results in the SC[37]. A cell-type-specific analysis that focuses on the adaptation rates of cortico-collicular neurons in L5 would provide valuable insight into understanding the circuit basis of transient responses in the SC. Brainstem-derived post-synaptic potentials in SC neurons rapidly adapt during 20 Hz stimulation, which is similar to the natural whisk frequency of mice[25]. Therefore, somatosensory adaptation in the SC could result from the loss of input excitation. However, a rapid stimulus filter built by longer-lasting inhibition is also plausible[45,50]. The duration and distribution of inhibition in the SC could regulate responses to gains in whisker contact, potentially through divisive normalization[51,52]. Interestingly, among the population of recorded SC neurons, we discovered a diversity of adaptation profiles whereby some neurons had a persistent self-generated response. The response dynamics of these neurons are more similar to the brainstem, thalamus and upper layers of the barrel cortex[37,53–58]. Therefore, multiple mechanisms of adaptation could co-exist in the SC that vary according to cell type or stimulus quality. Much work remains in understanding the functional diversity of cell types in the SC[59]. Future work revealing the adaptation rates of SC neurons in the context of their downstream connections would provide critical

insight into understanding the role of adaptation to sensory-guided movements and attention[10,60–62].

Neuronal selectivity for externally generated stimuli has been observed in other sensory modalities and brain areas. Neurons in the monkey cerebellum selectively respond to external but not self-generated vestibular accelerations[63]. Neurons in the monkey visual cortex display a unique window of suppression only when voluntary eye movements create visual stimulation[64]. Neurons in the superficial visual cortex of mice selectively respond to mismatches between visual flow and locomotion[65,66]. Neurons in the auditory brainstem cancel self-generated sounds associated with licking[67], and neurons in the hindbrain of weakly electric fish cancel self-generated electric signals[68]. We revealed a class of neurons in the superior colliculus that rapidly adapts to self-generated tactile stimulation. This cancellation of self-generated touch was specific to individual whiskers. Therefore, tactile adaptation in the SC follows a somatotopically organized 'labeled line'. At first glance, this may appear to be a limitation of the system. However, the natural density of the whisker array means that external movements are likely to gain contact with non-adapted whiskers, especially when combined with self-motion. One limitation of our study is that we did not have predetermined control over which whiskers gained or lost contact with the surface during external motion. Such a predetermined design would enable an understanding of how specific combinations of multi-whisker stimuli (GoW and LoW) summate in SC neurons.

What circuits are necessary for attenuating the self-generated tactile response? An efferent copy of whisker movement involving the cerebellum, motor cortex and brainstem could be an important mechanism for building the sensorimotor predictions that control transient response magnitude[13,67,69–73]. Several lines of evidence point to the cerebellum as a critical node in this process. In our study, sensorimotor predictions built by free-whisking were critical for enhancing the externally generated tactile response. In essence, a significantly larger tactile response emerged when the self-generated stimulus history did not match the external change in tactile space. Therefore, one function of the transient response may be to guide corrective movements during the pursuit of a moving target[13].

Habituation of the transient response may be a neural mechanism for encoding the novelty or expected consequence of environmental cues[74–80]. Since mice in our study behaved voluntarily and were not seeking reward, SC habituation may represent a central mechanism for ignoring stimuli of diminishing novelty[45,81]. Whether habituation relies on changes in inhibitory signaling from the substantia nigra, the activity of local SC interneurons, or other upstream circuits remains unknown[6,77,82]. Rapid habituation to repetitive visual stimulation has been observed previously in primates and rodents[83–85], yet our study demonstrates a slower, accumulative effect. Since GoW stimuli were repeated at 1-min intervals over the course of almost an hour, SC neurons appear to be updating sensorimotor predictions for an extended period.

The primary function of the transient response may be to facilitate the localization of object motion during active sensing. We show that population-level firing rates briefly, yet accurately, encode the location of external motion. The net effect of this brevity is a code for motion direction. Population decoding of external motion was most accurate between movements that engaged different whiskers. Therefore, transient responsive neurons in the SC may be specialized for encoding the leading edge of object motion through the whisker pad. The dynamics of spiking distributed across somatotopic space, generated by consecutive gains in whisker contact, as demonstrated in our study, could enable the orienting movements necessary for prey capture[20,86].

To localize objects using active touch, animals must differentiate real-world object motion from self-generated changes in tactile space. Such computations are pervasive across different sensory modalities and brain areas involved in sensorimotor processing. In the SC, response selectivity for unexpected, externally generated touch is controlled by rapid, whisker-specific adaptation. Sensorimotor predictions built by active sensing and repetitions in external motion controlled the magnitude of the transient response. Therefore, SC neurons contextualize tactile information within self- and externally generated stimulus histories across multiple timescales. It is perhaps surprising that single neurons in an ancient midbrain structure multiplex information across these dimensions. Yet, given the diversity of sensory, motor, and associative circuits that target the mammalian SC, this is also reasonable. Precisely how each of the different inputs to the SC shapes its dynamic sensorimotor maps and guide animal behavior remains an exciting and important topic.

## Methods

### Experimental model and subject details

Mice of a CD-1 background of both sexes, between the ages of 9 and 15 weeks, were used for all experiments. The Purdue Institutional Animal Care and Use Committee (IACUC, 1801001676A004) and the Laboratory Animal Program (LAP) approved all procedures. Mice were housed at room temperatures ranging between 68 and 79°F with humidity ranging between 40 and 60%. Mice were socially housed with five or less per cage and maintained in a reverse light-dark cycle (12:12 h). All experiments were conducted during the animal's subjective night.

### In-vivo electrophysiology

Prior to behavioral conditioning, a custom aluminum headplate was attached to each mouse to enable head fixation. Briefly, animals were anesthetized under 5% isoflurane and maintained at ~3% while monitoring body temperature and respiratory rate throughout the procedure. Artificial tears ointment was applied to keep the eyes hydrated. The skin and fur over the skull were disinfected with 70% ethanol, followed by betadine and incised using sterilized surgical instruments. A tissue adhesive (Liquivet) and dental cement (Metabond) were applied over the skull and wound margins. The headplate was attached to the skull with dental cement. Lastly buprenorphine (mg/kg) was administered as a postoperative analgesic. Two days after headplate implantation, mice were placed on a circular treadmill for 1 h per day for up to 10 days or until they learned to run freely at a steady pace.

On the day of the electrophysiology recording, mice were briefly (15–20 min) anesthetized to perform a craniotomy over the SC. A 1 mm diameter craniotomy was made using a biopsy punch (Robbins Instruments) and then covered with a silicone gel (Dowsil). Several hours later, mice were placed in the experimental rig, and a three-shank custom probe (Neuronexus) of 128 channels was lowered into the brain using a micromanipulator (NewScale). After exiting the ventral surface of the cortex, as evident by a loss of spiking, the electrode was lowered at a rate of 75 μm/min while constantly searching for activity driven by flashes of light. The onset of visual activity was used to mark the depth of the SC surface (~1000 μm below the cortical surface). The electrode was further lowered into the intermediate and deep layers, where manual whisker deflections caused strong neuronal responses. The receptive field of neurons was mapped by manually deflecting individual whiskers and locating the primary drivers of neuronal activity. Whiskers that did not elicit detectable activity were trimmed. Recordings were targeted toward the C-row and macro-vibrissae. If the electrode penetration missed the target, it was removed and re-inserted based on the coordinates of somatotopic space. In many experiments, mice had 3–5 intact whiskers contacting the surface, which spanned one or two rows. In other cases, mice had one or two whiskers intact.

### Whisker-guided virtual navigation

Head-fixed mice ran at their own volition (23.7 ± 1.6 cm/s, 12 mice) with concomitant rhythmic whisking (19 ± 1 Hz, 10 mice). All data presented

was collected during locomotion and whisking unless otherwise stated. The circular treadmill was attached to a digital encoder that controlled the rotation of a tactile surface in a closed loop. A trial began after 200 cm of locomotion on the treadmill to ensure the mice were actively sensing the surface. Shortly (1.4 s) after whisker imaging began, the surface translated 1 cm linear distance into rostral or caudal space or remained at the center location. Each outcome was randomly chosen with equal (33.33%) probability. In six mice, an additional location was added that was entirely outside the whisker field, giving each surface movement a 25% probability. While at the rostral/caudal position, mice had to run an additional 200 cm before the surface would return to the center. An opaque flag over the eye and white background noise obscured visual and auditory cues, respectively.

In two mice, the tactile surface was replaced with a pneumatically controlled rectangular surface. Prior to the experiment, all whiskers were trimmed except the principal whisker (B1 or C1). A trial began after the mouse ran 200 cm on the treadmill while freely whisking in air with no surface in the whisking field. After the mouse crossed the distance threshold and stopped running, the touch surface was extended into the whisking field. The mouse voluntarily resumed whisking and generated an unexpected touch event. The mouse then voluntarily stopped whisking for a period before resuming to touch the same surface and generate an expected touch event. After running 600 cm with the surface present, the surface retracted, and the trial restarted.

### Whisker tracking & kinematics

Whiskers were tracked at 500 fps during the trial. A high-speed infrared camera (Photonfocus DR1) and a mirror angled at 45° captured whisker motion under IR illumination. Videos were synchronized with neural data via external triggers generated by a National Instruments card and recorded on an Intan 512 controller. DeepLabCut was used to label the whisker(s) and track their movement[35]. Four evenly spaced points on each whisker were labeled. Approximately 150 frames were manually labeled from each experiment spanning all surface positions. The DLC neural network was trained for at least 200k iterations, and the final labels were manually checked for accuracy. Whisker position/angle was calculated for each label on each whisker with reference to a user-defined point on the face relative to the frame's vertical axis. Whisker angle was bandpass filtered (1–30 Hz, fdesign.bandpass order 4, MATLAB). Whisker bend/curvature was calculated from the three distal labels on each whisker using Menger curvature. The whisker curvature derivative was calculated as the local slope with cubic approximation in a moving window of 100 ms.

### Spike sorting

Spikes sorting was performed using the Kilosort2[87] and manually curated using Phy2 gui (https://github.com/cortex-lab/phy). Spike clusters were considered single units based on their spike waveform, location on the electrode, and cross-correlogram of spike times. Single units were used for all analyses in the paper.

### Statistical analyses

**Surface movement statistical classifier.** To classify the location of the tactile surface from population-level SC activity, we applied the Neural Decoding Toolbox[88] using a support vector machine (SVM). The classifier predicted surface location (rostral, caudal, center) throughout the duration of the trial. Only neurons with a significant response to surface translation were used. Spike data was first z-scored to prevent high firing rate neurons from having a disproportionate influence on classification. The spike rate of every neuron was calculated using 150 ms bins, and the surface location was predicted every 20 ms to plot classification accuracies over time. A 10-fold cross-validation was performed along with 50 resample runs to calculate a robust estimate of classification accuracies. The classification accuracy was measured based on the zero-one loss function.

**Modulation indices.** The gain and loss of whisker modulation indices (GoW/LoW indices) were calculated as the difference divided by the sum of spike rate averages that were calculated 500 ms before and after the start of surface movement.

**Transient and sustained neuron classification.** Surface movement was defined as a GoW/LoW stimulus when at least one whisker gained or lost contact. Since changes in whisker contact (GoW/LoW) occurred at slight time differences relative to the onset of surface movement across mice, we selected a custom transient window for each mouse based on the population average firing rates for every GoW/LoW translation. The peak firing rate of the transient response was identified by eye, and the troughs around the peaks were manually marked as the start and the end of the transient window. We verified this window onset with an analysis of whisker kinematics (whisker curvature and position). The firing rate of the sustained response was calculated in a 300 ms window that started one second after the transient window ended. We performed a one-way ANOVA (MATLAB anova1) on the firing rates calculated in the baseline (300 ms pre-movement), transient, and sustained windows for every neuron. A Tukey post hoc test (multcompare, MATLAB) was used to test for significant ($\alpha < 0.05$) differences, correcting for multiple comparisons. Neurons that had significantly different baseline and sustained firing rates were classified as sustained neurons. Neurons with a transient window firing rate that was greater than both their baseline and sustain window firing rates were classified as transient neurons. Neurons that had a significant transient and/or sustained response were considered tactile responsive and used for analysis.

**Self-GoW and external-GoW (Fig. 3).** Self-GoW and external-GoW firing rates were calculated in a 50 ms window following touch onset. Self-GoW was calculated as a difference between mean firing rates during self-GoW and self-motion. Self-motion occurred when the animal transitioned from resting to locomotion but free-whisking in air. The external-GoW response was calculated as the mean firing rate in the touch window minus the pre-touch (free-whisking) window.

**Expected and unexpected touch (Fig. 4).** Expected and unexpected touch firing rates were calculated by subtracting pre-touch firing rates from post-touch in a 50-ms window.

**Stimulus repetition.** To obtain repetition slopes, we fit the firing rates of transient neurons in the transient window and sustained neurons in the sustained window with a linear regression model using the MATLAB fitlm function. We only used neurons that did not drift. Neuronal drift was determined by testing if the baseline firing rate during the first 10 trials was different from the last 10 trials using a Wilcoxon signed rank test. Spike amplitudes were obtained from 'amplitudes.npy' output from Phy GUI. Normalized whisker positions across repetitions were obtained as the mean of the normalized whisker position in the transient response window for each repetition. Spike waveform amplitudes across repetitions were obtained by taking the mean of the average normalized waveform from all spikes in the transient window of each repetition. A scaling to range normalization method was performed as below.

$$Xnorm = \frac{X - X\min}{X\max - X\min} \tag{1}$$

### Reporting summary

Further information on research design is available in the Nature Portfolio Reporting Summary linked to this article.

## Data availability

The data that support the findings of this study have been deposited in the DANDI database under accession code 0.230806.0034[89]. Source data are provided with this paper.

## Code availability

All code is written in MATLAB and is available from the authors upon request.

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

## Acknowledgements

The authors would like to acknowledge the members of the Pluta lab, Julia Veit, Kate Hong, Edward Bartlett, and Daniel Butts for providing valuable feedback on the manuscript. This work was supported by the Whitehall Foundation, the Showalter Trust, and the Purdue Institute for Integrated Neuroscience to S.R.P.

## Author contributions

S.R.P. and S.C. designed the experiments. S.C. performed the experiments and analyzed the data. S.R.P. and S.C. wrote the paper.

## Competing interests

The authors declare no competing interests.
