## [Peer Review File · Nature Communications]

Neural mechanisms for the localization of unexpected external motionREVIEWER COMMENTS

Reviewer #1 (Remarks to the Author):

A key question in sensory physiology is how does an animal account for its own movement when conducting tasks such as detection. In the present article manuscript recordings are obtained from the superior colliculus while a head fixed mouse runs on a turn table and uses its whisker to sense stimuli put in its path. The whisker evoked responses to active touches were larger than those due to passive encounters when the mouse's whiskers essentially ran into the stimulus. Population activity correlated with the direction of external movement.

Overall, the study was of high quality with clear figures.

Issues

1. When the stimuli position was moved – how quickly was this accomplished – was this done in a way to ensure the mouse was not contacting it during its movement? Was the movement masked (white noise) so as to not cue the animal?
2. Trials were initiated after the mouse ran a specific distance. What was the threshold for this, and if the trial was delayed (more distance covered by the animal) did that impact the magnitude of the evoked response? Did the velocity of movement impact response magnitude?
3. Did the response magnitude in the Gain of Whisker versus Loss of Whisker (Figure 2) experiments vary as a function of the number of whiskers engaged (or disengaged) or the velocity of the whiskers when contact was gained or lost? Gain of whisker are loss of whisker responses are reminiscent of On/Off responses to whisker deflection at other levels of the neural axis – is this an apt comparison?
4. Line 72 'the firing rate of most neurons rapidly increased while others decreased' can n's or percent of neurons demonstrating these behaviors be included.
5. Were the larger responses seen with Gain of Whisker functions due to novelty or just larger afferent drive?
6. Were recording locations verified post-hoc?

Reviewer #2 (Remarks to the Author):

In this study the authors investigate neuronal responses in the superior colliculus (SC) in mice placed in a whisker-guided virtual reality. They found that neurons in SC strongly prefer external over self-generated changes in tactile space. In addition, a transient response only emerged in cases where the external stimulus gained contact with a whisker, implying a whisker-specific adaptation in the SC. Understanding how one can dissociate self and external movement is a timely and interesting topic. I also agree that there are too few studies in the SC of a behaving animal. This is especially true for the mouse whisker system, which can be nicely paralleled to visuospatial processing. Nevertheless, I have some major concerns that dampen my enthusiasm regarding this study:

1. The relationship between the experimental evidence and the main take home message (i.e., SC encodes externally generated tactile motion) is relatively weak. The only straight-forward evidence that SC neurons encode external stimulus motion as compared to self-generated motion is presented in Figure 3 in a specific experiment. The rest of the results either set the ground (Figures 1 and 2 introducing GoW and LoW touches) or control (Figure 4 and 5) for the results presented in Figure 3 (figure 6 is also unrelated directly to external motion). Having narrowing down the results to one figure, the results in figure 3 can be explained by other factors that are not necessarily external motion:

- Initial whisker state/position of external and self-motion are distinctly different. In the self-motion case, the whiskers are in a retracted position and actively protract. In the external case, the whisker seems to be in an already protractive state when the surface comes into contact and then even retract. This difference can be also seen in the whisker curvature just before touch in the two conditions (Fig 3D and S2, green higher than red). In this context, the differences in SC responses could be related to the fact that in the self-motion case the mouse actively whisked and activity in SC is already high and therefore the touch response is naturally lower. In other words, if we look at pre-touch phase, there seems to be a clear difference in whisker curvature (green higher than red in Figure 3D). This phenomenon occurring before the touch itself can directly effect responses in SC to upcoming touch. Would be nice to see what is the raw firing baseline in both conditions (i.e., not in z-score).

- As the authors point out, there is also a big difference in locomotion between the external and self-motion conditions. The authors directly relate to this issue in Figure 4 where they designed a dedicated experiment showing that unexpected touches display higher SC responses compared to expected touches (here, both conditions are controlled for locomotion). Well, this is indeed interesting, but does not directly compare external and self-initiated motion. These results also bring into question that expectation plays a role in SC responses which is not necessarily external motion (see my next major point). In sum, locomotion differences that may explain the main results in figure 3 are only partially addressed in Figure 4.

- Why don't the authors find this type of evidence in the experiment presented in figure 2? In that experiment the surface has several different positions, so one could maybe compare a wider range of whisker positions and other parameters. Do the authors see difference in self and external GoWs? Reading this manuscript, I had some conceptual gaps between the different experiments.

In summary, I felt there are too little results which were not well-controlled causing in a weak consolidation for the main conclusions of the manuscript.

2. Continuing the previous point, as I was reading the manuscript, I had a mixed-up feeling of terminology such as external/self-motion, expectation, whisker-specific adaptation, long-lasting habituation, prediction error. All of these terms were at some point brought into the text and from my viewpoint just raised more questions. For example, Figure 4 shows that SC neurons encode expectation. But how is expectation linked to self/external motion (this is not well linked in the text)? Is it maybe that external motion is more unexpected? If so, then maybe an expected external movement will display a weaker response? Maybe the authors can design an experiment addressing this point? Another example, Figure 5 shows the external motion GoW display habituation after minutes. How is long-lasting habituation related to external motion? In other words, how does figure 5 serve figure 3 (the main result)? The mix up continues when relating to transient and sustained responses that appear in Figure 2 and then reappear in Figure 6. In contrast, sustained responses are not shown for external vs self-motion in Figure 3. So how do these types of responses help put forward the main take home message? In other words, how do figures 2 and 6 serve figure 3? In general, it seems that the manuscript presents lots of small pieces of the puzzle (each interesting by itself and each coming from a different experiment) but at the moment the pieces are far apart, not enabling to see a clear picture.

3. Even if we put aside the above major concerns, I feel that the basic results of higher SC responses for external motion is too simplistic. The authors should add some additional evidence (maybe causal, maybe different experiment) to support their claim. Even additional analysis into suppressive/enhancive neurons presented in Figure 2 or latency analysis (could self generated touch have an earlier SC response?) or trying to directly decode external and self motion. What happens if mice need to behaviorally report one position? Then maybe external stimuli will now be more expected and show lower response in SC? Maybe doing similar experiments when mice are not locomoting? Maybe having an auditory cue before texture comes in? In sum, there should be a substantial addition of evidence supporting the main conclusions of the manuscript.

Minor points:

- In figure 3 the authors should clearly present the locomotion of the mouse, since this is an important aspect in the experiment
- Could other factors such as body movements or pupil dilation effect the results?
- The discussion is missing a discussion on why they find higher response in SC for external motion. In general, the discussion reads a bit fragmented, trying to address many previous terminologies (see major point 2), but not telling a coherent story
- Almost each figure had a different color code. Sometimes green and red, sometimes black and red, sometimes several other colors. I had to put substantial effort into following each figure separately.

- I had an issue with embracing transient and sustained responses. I could not grasp how it fits the whole story.

Reviewer #3 (Remarks to the Author):

In their paper 'Neural mechanisms for the localization of externally generated tactile motion' the authors describe collicular somatosensory responses in awake behaving mice. At the outset, the authors note that little is known about tactile responses in the superior colliculus and this is very much true. The authors conduct their experiments in, what they call a tactile virtual reality. Whisker contacts are monitored with high-speed videography. The authors use multiple multi-contact electrodes and record a substantial number of somatosensory collicular neurons. In the experimental conditions chosen by the author it is challenging to verify tactile contacts and responses and the authors do a great job in resolving these issues. They first document tactile responses related to whisker contacts. The core finding of the authors is that collicular neurons respond much more strongly to externally imposed stimuli than to self-generated contacts. This is a difficult issue to address, but the authors' data are quite convincing. The authors also provide data that expectation and stimulus repetition greatly affect collicular response amplitudes. My overall assessment of the paper is quite positive. It is true that we know little about these neurons and the careful analyses of this paper will change the status quo positively.

My queries are minor. First, I think the authors could do a better job in characterizing whisking in their virtual reality. The high whisking frequencies seen in this setting are quite striking, but they are barely mentioned by the authors. Also there seem to be quite marked changes in whisking frequency and parameters across the author's behavioral paradigm, but again they are not well quantified. What also limits the impact of the paper is that the key response aspects quantified by the authors (self- vs non-self responses, expectancy, adaptation) are barely compared to somatosensory responses in other brain regions (barrel cortex, trigeminal cells etc). The paper needs to devote more effort to such comparisons, ideally by conducting a subset of measurements in other brain regions in their experimental paradigm or by a more expansive comparison to existing data.

We are grateful to the reviewers for their valuable feedback on our manuscript. We have spent considerable effort addressing their concerns to the best of our ability. In summary, we made extensive changes to Figure 2, which suffered from too many distractions. We have added four high-speed video clips that correspond to the data presented in Figure 2A. We updated figure 1 to focus more on the transient response. We added running information to Figure 3 and additional supplemental analyses that address concerns related to touch conditions and pre-touch firing rates. We have added new supplemental figures that address concerns related to whisk frequency, multi-whisker integration, and the speed of surface translation. We vastly improved the narrative around Figure 4, making it part of the same cohesive message. Moreover, we greatly streamlined the narrative throughout the manuscript and improved its overall consistency. We made color schemes more consistent in figures and we updated Figure 6 to remove distractions: focusing on the external-selective, transient responsive neurons. Lastly, we improved the discussion by addressing each major finding in our paper and discussing it in the context of previous work, in addition to addressing the limitations of our study.

Overall, the study was of high quality with clear figures.

We are happy that you find the data and figures of high quality. Thank you.

Issues

1. When the stimuli position was moved – how quickly was this accomplished – was this done in a way to ensure the mouse was not contacting it during its movement?

Surface movement into a new location took approximately 175 ms. At all three locations (central, rostral, and caudal), the mouse was rhythmically whisking against the surface with an inter-touch-interval of approximately 60 ms. Therefore, during the surface translation, the mouse was continually contacting the moving object. As discussed in Figure 2 of results, many object translations from the center to the rostral/caudal location gained contact with a single whisker. The other whisker(s) sustained rhythmic contact with the surface. In a subset of experiments, the surface moved rostrally beyond the whisking field, losing contact with all whiskers. We carefully analyzed whisker kinematics to determine precisely when gains and losses in whisker contact occurred. We identified a rapidly adapting “transient” response in SC neurons that only emerged during surface movements that gained/lost contact with a whisker. As shown in Figure 2A and Suppl. Figure 2A,B and F surface movements that sustained contact with the same whiskers did not generate a transient response. This means that externally generated gains in whisker contact caused the transient response, which then rapidly adapted, despite the persistence of active (self-generated) rhythmic touch against the surface. Given this knowledge, our next experiment tested whether self-generated gains in whisker contact would evoke a similar transient response. To test this hypothesis, we performed experiments in figure 3 and 4.

In Figure 3, the mouse touched the surface near the end of surface movement. We carefully selected trials where the mouse’s volitional movement caused contact with the moving surface, removing trials where the surface bumped into the whisker when it was in a protracted state. We carefully analyzed the curvature of the whisker, comparing the quality of externally- and self-generated touch. As shown in supplemental Figure 5, touch quality between the conditions was remarkably similar across all four animals. Moreover, the angle of whisker contact had no effect on response magnitude (Suppl. Fig. 5C). Nonetheless, we wanted to be rigorous, and therefore we devised a new experiment, shown in Figure 4.

Here, touch conditions were identical on every level, yet most neurons strongly preferred GoW stimuli that were induced by an external change in tactile space. For figures 3 and 4, some neurons had identical responses to externally and self-generated GoW stimuli, providing further evidence for similarity in touch quality across conditions.

Was the movement masked (white noise) so as to not cue the animal?

As mentioned in paragraph 1 of results, movement-associated sounds were masked with white noise and an opaque flag was placed over the eye to prevent visual cues.

2. Trials were initiated after the mouse ran a specific distance. What was the threshold for this, and if the trial was delayed (more distance covered by the animal) did that impact the magnitude of the evoked response? Did the velocity of movement impact response magnitude?

A new trial started after the animal ran 200 cm. We did not vary this distance, as that was beyond the scope of this study. In a few experiments (not included in the primary analysis), we did use two different surface movement speeds. This had no impact on neural response magnitude – only a shift in the onset of the response, due to the surface gaining contact earlier in time. We have added data from two animals showing no effect of surface movement speed on transient response magnitude. This is likely because the mouse is actively whisking, and every touch is above the force threshold (Suppl. Fig 3). One factor that controlled the magnitude of the transient response was stimulus repetition, as shown in Figure 5.

3. Did the response magnitude in the Gain of Whisker versus Loss of Whisker (Figure 2) experiments vary as a function of the number of whiskers engaged (or disengaged) or the velocity of the whiskers when contact was gained or lost?

Thank you for this question. We have now added supplemental data illustrating this effect (Suppl. Fig 4). Overall, response magnitude during GoW stimulation was not affected by the number of whiskers involved. However, we imagine that multi-whisker summation does occur for specific combinations of whiskers. The duration of the response increased for some neurons because as additional whiskers gained contact additional spiking occurred. This effect was more notable during LoW stimulation. Importantly, we have changed the examples in figure 2, so now multi-whisker touch is no longer displayed, as we feel this is largely a distraction from the main point of the paper.

Some neurons were suppressed by whisker touch. These neurons had a larger increase in firing rate during complete loss of touch. So, there is likely some additive effect of touch suppression, but this result is simply beyond the scope of this paper.

As discussed above, the velocity of object movement had no effect on tactile responses. We estimate this occurred due to our stimulus being well-above the response threshold, thereby evoking a near maximal response on every trial. In addition, because the mice were running at a steady pace, they were whisking in a very stereotyped manner that was consistent across trials. This was by design.

Gain of whisker and loss of whisker responses are reminiscent of On/Off responses to whisker deflection at other levels of the neural axis – is this an apt comparison?

251/459 tactile responsive neurons had a GoW and a LoW response. I think it is reasonable to compare these dynamics to the increase and decrease in follicle tension caused by a passive deflection and release of the whisker. We have updated the results to highlight this similarity. Touch suppressed neurons that displayed a LoW response are possibly controlled by a different mechanism that involves disinhibition.

4. Line 72 'the firing rate of most neurons rapidly increased while others decreased' can n's or percent of neurons demonstrating these behaviors be included.

This sentence has been removed because it is largely a distraction from the main point. We have a new supplemental figure 2 that shows how many neurons were facilitated/suppressed by external-GoW touch.

5. Were the larger responses seen with Gain of Whisker functions due to novelty or just larger afferent drive?

In the context of figure 2 and supplementary figure 2, we think GoW responses being larger than LoW responses is due to afferent drive (force differentials on the follicle). Since the direction of surface movement was randomly chosen, GoW and LoW stimuli did not have differences in novelty.

6. Were recording locations verified post-hoc?

Yes, in most cases. In all cases, we carefully mapped the receptive field of neurons by hand to ensure that we were in the macrovibrissae region of the SC. We used visual stimulation to track our progress through the visual cortex and the superficial layers of the SC.

Reviewer #2 (Remarks to the Author):

In this study the authors investigate neuronal responses in the superior colliculus (SC) in mice placed in a whisker-guided virtual reality. They found that neurons in SC strongly prefer external over self-generated changes in tactile space. In addition, a transient response only emerged in cases where the external stimulus gained contact with a whisker, implying a whisker-specific adaptation in the SC. Understanding how one can dissociate self and external movement is a timely and interesting topic. I also agree that there are too few studies in the SC of a behaving animal. This is especially true for the mouse whisker system, which can be nicely paralleled to visuospatial processing. Nevertheless, I have some major concerns that dampen my enthusiasm regarding this study:

We are glad that you find this topic timely and interesting. We are grateful that you have expressed in detail your concerns with the paper, as these thoughtful comments are necessary for making this manuscript better. Your comments have undoubtedly improved the quality of our manuscript. We hope that we have adequately addressed your concerns.

1. The relationship between the experimental evidence and the main take home message (i.e., SC encodes externally generated tactile motion) is relatively weak. The only straight-forward evidence that

SC neurons encode external stimulus motion as compared to self-generated motion is presented in Figure 3 in a specific experiment.

Thank you for enumerating your concern. However, this is simply untrue. Nonetheless, we believe this is a great opportunity for us to improve the narrative, make the results more transparent, and remove unnecessary distractions, so this misunderstanding does not occur. We have updated figure 2 to improve the narrative.

Now, the main take-home from figure 2 is that the transient response only emerges when externally generated movements engage a new whisker. Throughout our entire experiment, the mice were constantly and rhythmically whisking against the surface (self-generating touch). Most SC neurons were simply unresponsive (approximately zero spikes/s) to these self-generated touches. However, when external motion contacted a NEW whisker (external-GoW), a rapidly adapting (transient) response emerged. At the end of this transient response (~250 ms later), the mouse continued to rhythmically whisk against the surface, but now with an additional whisker contacting the surface. Once again, neurons displayed approximately zero spikes per second. In short, self-touch = no spikes; external touch = transient burst of spikes. This short burst of activity is nearly identical to SC response dynamics during visual motion presentation to a cat, monkey, or mouse (Waleszczyk et al., 2004; Moors and Vendrik, 1979; Gale and Murphy, 2016). In essence, the rhythmic, self-generated whisker touches in mouse is analogous to a visual fixation on a stationary object. It has been published throughout SC literature that many SC neurons (especially in peripheral space) only respond to visual motion, and not stationary objects. This literature has been added to the introduction to help motivate our study for the reader.

The rest of the results either set the ground (Figures 1 and 2 introducing GoW and LoW touches) or control (Figure 4 and 5) for the results presented in Figure 3 (figure 6 is also unrelated directly to external motion).

Figure 5 is in no way a control. It demonstrates that stimulus history (acquired via external stimulus repetition) controls the magnitude of the externally generated transient response. We have updated our results to make this point clearer.

Figure 6 is directly related to external motion, since we focus our analysis on the transient responses that only emerged during external motion. We can see how this is confusing to the reader, however. Therefore, we have updated figure 6 to remove focus from the neurons that have a sustained response – the subset of neurons that persistently responded to self-generated touch.

Figure 4 is a critical control that stands on its own merit that strongly supports the result of Figure 3. Since studying active touch has a unique set of challenges, we believe this control is necessary to remove any shred of doubt that may arise from figure 3.

Having narrowing down the results to one figure, the results in figure 3 can be explained by other factors that are not necessarily external motion:

- Initial whisker state/position of external and self-motion are distinctly different. In the self-motion case, the whiskers are in a retracted position and actively protracts.

Despite their initial position, the whisker touched the object at nearly the exact same location across external- and self-GoW conditions. However, because the location was not exactly the same, we devised the experiment in Figure 4.

In addition, we have added a new analysis to Supplemental Figure 5C that demonstrates that the angle of contact has no effect on external-GoW response magnitude. This proves that the small differences in the angle of touch cannot explain the massive response difference between external and self-GoW stimulation.

In the external case, the whisker seems to be in an already protractive state when the surface comes into contact and then even retract.

In both cases, the whisker protracted, contacted the object, and then retracted, since the mice were whisking rhythmically. We carefully selected external-GoW trials where volitional touch was initiated by whisker protraction. We removed the small subset of external-GoW trials where the object simply pushed against the whisker in its protracted state. This point has been added to the results.

This difference can be also seen in the whisker curvature just before touch in the two conditions (Fig 3D and S2, green higher than red).

If you look at the data from individual mice shown in supplemental figure 5A, curvature at the moment of touch is nearly identical. We have plotted raw firing rates during the pre-touch period in Suppl. Fig. 5D.

In this context, the differences in SC responses could be related to the fact that in the self-motion case the mouse actively whisked and activity in SC is already high and therefore the touch response is naturally lower.

The mouse is actively whisking in both the self- and external-GoW conditions, as shown in the whisker traces of Figure 3.

In other words, if we look at pre-touch phase, there seems to be a clear difference in whisker curvature (green higher than red in Figure 3D). This phenomenon occurring before the touch itself can directly effect responses in SC to upcoming touch. Would be nice to see what is the raw firing baseline in both conditions (i.e., not in z-score).

We have now added raw firing rates for the pre-touch and the touch periods in Suppl. Fig. 5D,E in addition to the raw firing rates observable in the three example neurons. There was no difference in pre-touch firing rates between the external- and self-GoW conditions (Suppl. Fig. 5D, Wilcoxon signed rank test $p = 0.81$), whereas the raw firing rates for external-GoW responses were significantly greater than self-GoW raw responses (Suppl. Fig. 5E, Wilcoxon signed rank test $p = 2e-10$). This proves that the pre-touch firing rates do not explain the larger external-GoW response.

As shown in Suppl. Fig. 5A, there were no differences in pre-touch curvature for 2 of the 4 mice, yet all four mice show the same preference for external-GoW.

- As the authors point out, there is also a big difference in locomotion between the external and self-motion conditions. The authors directly relate to this issue in Figure 4 where they designed a dedicated experiment showing that unexpected touches display higher SC responses compared to expected touches (here, both conditions are controlled for locomotion). Well, this is indeed interesting, but does not directly compare external and self-initiated motion.

Thank you for making clear where the narrative needs to change. Figure 4 does in fact compare externally and self-generated stimulation. I will use an analogy to the visual system since it is more intuitive to understand. Imagine you are looking straight ahead into empty space. You then close your eyes for 1 second, and during this period an object moves into your visual field. When you open your eyes, your visual system responds to this externally generated (unexpected) change in the visual scene. The object then stays at that exact location, and you continue to look at the object, but eventually you close your eyes. When open your eyes again, you see the same object in the same location. You repeat this sequence 10 times. Every time you open your eyes, your visual system responds to that persistent object. The repeated visual response to opening your eyes (self-generated) is smaller than the visual response that occurred the moment after the object moved (externally generated) into the scene. This difference is caused by a mismatch in stimulus history: visualizing an empty space before the object vs. repeatedly visualizing the same object.

In our experiment, when the mouse was actively whisking in air, it was “visualizing” an empty space. After the mouse momentarily stopped whisking in that space, an object moved into it. When the mouse resumed whisking into that space, a touch occurred that was caused by [unexpected] external motion (mice had no way of predicting when/if an object would enter its whisking field). Without this object movement, there was no self-generated movement that could have generated this sensory experience. Therefore, the object entering its whisking field was in direct violation of the sensorimotor predictions generated by active whisking in air. This logic is identical to the main experimental system (figures 1, 2, 3, 5, and 6).

These results also bring into question that expectation plays a role in SC responses which is not necessarily external motion (see my next major point). In sum, locomotion differences that may explain the main results in figure 3 are only partially addressed in Figure 4.

Figures 3 and 4 test the same hypothesis and figure 4 is a necessary control that addresses any potential concerns in figure 3. I hope our change in narrative makes that clear.

During active sensing, external motion is perceived when sensorimotor predictions generated by self-motion do not match actual sensory feedback.

The main point of figures 3 and 4 is that expectations, built by self-generated stimulus history, control the magnitude of the transient response. Sensorimotor predictions directly conflict with unexpected/externally generated stimuli, which is what our experiment tested. Our experiment mimicked natural exploration where no prior expectations about the environment existed (mice were seeing the experimental system for the first time in their life).

Self-generated touch is perceived fundamentally differently than externally generated touch. During active sensing, an efferent copy of self-motion is sent to the sensory system to generate the expectation/prediction of self-induced stimulation. In our experiment, when the object moved, it contacted a new whisker that self-generated movements did not contact and therefore could not “predict”. In conclusion, when the object moved in our experiment, it violated the expectations built by self-generated stimulus history: all the self-generated touches that occurred prior to an external change. This logic is identical to the work from George Keller’s lab in the visual cortex of mice (Zmarz & Keller 2016).

- Why don't the authors find this type of evidence in the experiment presented in figure 2?

We do see evidence for expectation from the experimental system presented in Figures 1, 2, 3, 5, and 6. This is the main point of figure 5, where accumulating stimulus history weakens the externally generated transient response. Figure 3 also shows the effect of [self-generated] stimulus history. The mouse starts the trial by whisking in air, and the predictions generated by this movement are violated when the object moved into its whisking field. When the mouse doesn't actively whisk (doesn't change stimulus predictions) between touch events (as the case in Figures 3 and 4), the tactile response is weaker. In Figure 2, object movements that maintained contact with the same whiskers did not generate a transient response, indicating that self-generated stimulus histories/predictions are specific to individual whiskers.

We have updated the figures and results to make these points clear.

In that experiment the surface has several different positions, so one could maybe compare a wider range of whisker positions and other parameters. Do the authors see difference in self and external GoWs? Reading this manuscript, I had some conceptual gaps between the different experiments. In summary, I felt there are too little results which were not well-controlled causing in a weak consolidation for the main conclusions of the manuscript.

We are sorry for this misunderstanding. Due to your careful insight, we now see why there is confusion, and we hope that changes we made to the text and figures better explain how data across multiple figures compares external and self-generated touch.

2. Continuing the previous point, as I was reading the manuscript, I had a mixed-up feeling of terminology such as external/self-motion, expectation, whisker-specific adaptation, long-lasting habituation, prediction error. All of these terms were at some point brought into the text and from my viewpoint just raised more questions. For example, Figure 4 shows that SC neurons encode expectation. But how is expectation linked to self/external motion (this is not well linked in the text)? Is it maybe that external motion is more unexpected?

Thank you for your feedback. Yes, externally generated touch with no prior history is less expected than volitional touch against a known object. We have updated the text in many locations to make this narrative clear.

If so, then maybe an expected external movement will display a weaker response?

Yes, we show this result in Figure 5. As the experiment progresses, the object movements become very repetitive/predictable since there are only four trajectories total and two of those trajectories are 100% predictable from the previous trajectory. This predictability with repetition is observed as habituation in the transient response.

Another example, Figure 5 shows the external motion GoW display habituation after minutes. How is long-lasting habituation related to external motion? In other words, how does figure 5 serve figure 3 (the main result)?

Figure 5 is an elaboration of the Figure 3 result, showing that stimulus history accumulates, ultimately controlling the magnitude of the transient external-GoW response.

The mix up continues when relating to transient and sustained responses that appear in Figure 2 and then reappear in Figure 6. In contrast, sustained responses are not shown for external vs self-motion in Figure 3. So how do these types of responses help put forward the main take home message? In other words, how do figures 2 and 6 serve figure 3?

Thank you for your input. Figure 6 now focuses on describing the object localization ability of the transient (external motion selective) responsive neurons.

3. Even if we put aside the above major concerns, I feel that the basic results of higher SC responses for external motion is too simplistic.

Externally generated object motion with no prior knowledge is an unexpected change. Previous work in other sensory modalities has shown that self-motion is suppressed allowing external motion stimuli to be amplified (Miura & Scanziani 2021; Rummell, Klee & Sigurdsson 2016; Singla, ..., Sawtell 2017). These works, including ours, agree with theories of predictive codes generated by self-motion. Our study is the first to show this neural computation during active whisking, although it has been hypothesized from carefully designed behavioral studies.

The authors should add some additional evidence (maybe causal, maybe different experiment) to support their claim.

Even additional analysis into suppressive/enhancive neurons presented in Figure 2 or latency analysis (could self generated touch have an earlier SC response?) or trying to directly decode external and self motion.

We show that neurons in the SC only respond to external motion. Therefore, they only decode external motion.

What happens if mice need to behaviorally report one position?

Reward associations influences SC activity independent of touch. That would simply add a confounding variable.

Then maybe external stimuli will now be more expected and show lower response in SC? Maybe doing similar experiments when mice are not locomoting? Maybe having an auditory cue before texture comes in? In sum, there should be a substantial addition of evidence supporting the main conclusions of the manuscript.

Sensorimotor predictions are naturally generated during self-motion and do not require sensory cues or reward.

Minor points:

- In figure 3 the authors should clearly present the locomotion of the mouse, since this is an important aspect in the experiment

Thank you for the suggestion. We have added locomotion speed to Figure 3.

- Could other factors such as body movements or pupil dilation effect the results?

Although SC activity could depend on pupil size, pupil dilation itself will not affect the results since the mouse's entire visual field was obscured.

- The discussion is missing a discussion on why they find higher response in SC for external motion. In general, the discussion reads a bit fragmented, trying to address many previous terminologies (see major point 2), but not telling a coherent story

We have done our best to improve the narrative. We now discuss how self-generated stimulation is known to be attenuated through an efferent copy of self-motion in multiple different brain areas and sensory modalities.

- Almost each figure had a different color code. Sometimes green and red, sometimes black and red, sometimes several other colors. I had to put substantial effort into following each figure separately.

We have made the colors more consistent. Thank you for the suggestion.

- I had an issue with embracing transient and sustained responses. I could not grasp how it fits the whole story.

Thank you for raising this concern. We have made considerable changes to the figures and all sections of text to focus on the transient response which is critical to the narrative of external motion selectivity.

Reviewer #3 (Remarks to the Author):

In their paper 'Neural mechanisms for the localization of externally generated tactile motion' the authors describe collicular somatosensory responses in awake behaving mice. At the outset, the authors note that little is known about tactile responses in the superior colliculus and this is very much true. The authors conduct their experiments in, what they call a tactile virtual reality. Whisker contacts are monitored with high-speed videography. The authors use multiple multi-contact electrodes and record substantial number of somatosensory collicular neurons. In the experimental conditions chosen by the author it is challenging to verify tactile contacts and responses and the authors do a great job in resolving these issues. They first document tactile responses related to whisker contacts. The core finding of the authors is that collicular neurons respond much more strongly to externally imposed stimuli than to self-generated contacts. This is a difficult issue to address, but the authors data are quite convincing. The authors also provide data that expectation and stimulus repetition greatly affect collicular response amplitudes. My overall assessment of the paper is quite positive. It is true that we know little about these neurons and the careful analyses of this paper will change the status quo positively.

Thank you for the positive feedback. We are happy that you find value in the work.

My queries are minor. First, I think the authors could do a better job in characterizing whisking in their virtual reality. The high whisking frequencies seen in this setting are quite striking, but they are barely mentioned by the authors.

The whisk frequencies occurring in this setting are the natural whisk frequencies produced by locomoting mice as reported by Voigts ...Celikel in 2008 (frequency $(19 \pm 7 \text{ Hz})$). The relationship between whisk frequency and locomotion was first documented by Sofroniew...Svoboda in 2014. It was further explored in the work of Pluta...Adesnik in 2017. We have also added an additional supplementary figure displaying whisking frequency profile during locomotion. (Suppl. Fig 1)

Also there seem to be quite marked changes in whisking frequency and parameters across the author's behavioral paradigm, but again they are not well quantified.

For the data we analyze and present, whisk frequency is remarkably stable, because we train our mice to run and we filter out trials where the mouse displayed a saltatory pattern (Suppl. Fig 1C). By using locomoting-only data, we control variation in whisking. For figures 3 and 4, where mice are stopping and starting locomotion, whisk frequency is not an important factor, because we only analyze spikes evoked by the first touch in each bout of whisking.

We have added a supplementary figure (Suppl. Fig 1C) that demonstrates the stability of whisk frequency across conditions.

What also limits the impact of the paper is that the key response aspects quantified by the authors (self vs non-self responses, expectancy, adaptation) are barely compared to somatosensory responses in other brain regions (barrel cortex, trigeminal cells etc). The paper needs to devote more effort to such comparisons, ideally by conducting a subset of measurements in other brain regions in their experimental paradigm or by a more expansive comparison to existing data.

The reviewer raised an interesting and thoughtful proposal comparing the results discovered in this article to other somatosensory brain regions. In line with this, our future research work is dedicated to exploring the concepts of how (i) expectancy, (ii) adaptation and (iii) habituation shapes neural responses in the somatosensory cortex, cerebellum, motor cortex, and the trigeminal nucleus. For the current article, we have expanded our discussion of how our data is similar to results from the barrel and visual cortices where similar works have been conducted.

REVIEWERS' COMMENTS

Reviewer #1 (Remarks to the Author):

the authors have effectively answered my concerns

Reviewer #2 (Remarks to the Author):

Thank you for the substantial revision of the manuscript. The authors have done a great job in clearly narrating the figures and adding more connecting sentences. The story now reads in a much more clear and logical way. The authors also tightened their use of different terminology which resulted in a coherent read. Most of my concerns were adequately addressed.

Some issues that I still have left.

The authors write in their rebuttal: "In essence, the rhythmic, self-generated whisker touches in mouse is analogous to a visual fixation on a stationary object." (this is actually a good sentence to put in the introduction)

This analogy has now been made clear to me, but also raised some questions. The main difference between fixational vision and active whisking is that the latter involves motor output not only in rhythmic whisking but also in body movements and locomotion. So unlike passive vision, active sensing is hard to disentangle into different parameters such as movement or external touch. For me, active whisking is more parallel to a visual search task where subjects are required to search with their eyes for a target within a noisy field.

This difference raises again figure 3 which is not well controlled for movement differences between elf and external touch. First, there are locomotion differences between conditions (as the authors mention in the text and now show in example figure 3). There seems to be a difference in whisker position before external and self touches. There still seems to be other baseline differences in movements which may have an effect. I definitely understand the complexity in this experiment and appreciate the additional experiment done to partially address this issue.

I think the authors should add a clear 1-2 sentences in the relevant results section where they directly state that there are other parameters such as locomotion or body movements that could explain these results. The authors should also quantify locomotion during self and external touch and present it clearly

in the main figure 3 (similar to Figure 3C, D and statistics). In addition, some discussion comparing fixational vision and active whisking would be nice.

Typo: Line 40: "... during sensor fixation". Should be sensory

Reviewer #3 (Remarks to the Author):

The authors addressed my concerns. I wish they did more on the comparison of the current results to responses in other brain regions, but I take their point that this is beyond the scope of the current study. Overall, this is an important paper, because data on somatosensory responses in the SC of awake rodents are missing.

Response to Referees Letter

Reviewer #1

the authors have effectively answered my concerns

We thank the reviewer for their valuable comments and suggestions.

Reviewer #2

The authors write in their rebuttal: "In essence, the rhythmic, self-generated whisker touches in mouse is analogous to a visual fixation on a stationary object." (this is actually a good sentence to put in the introduction)

This analogy has now been made clear to me, but also raised some questions. The main difference between fixational vision and active whisking is that the latter involves motor output not only in rhythmic whisking but also in body movements and locomotion. So unlike passive vision, active sensing is hard to disentangle into different parameters such as movement or external touch. For me, active whisking is more parallel to a visual search task where subjects are required to search with their eyes for a target within a noisy field.

We agree with the reviewer that active whisking involves body movement. Therefore, we used the word "analogous" to point out their similarities rather than "identical", since somatosensation and vision are inherently different. Visual search is again a great analogy to active whisking. Since touch in our experiment only occurred during movement, the effect of movement on transient response magnitude was reduced or minimal.

This difference raises again figure 3 which is not well controlled for movement differences between self and external touch. First, there are locomotion differences between conditions (as the authors mention in the text and now show in example figure 3). There seems to be a difference in whisker position before external and self touches. There still seems to be other baseline differences in movements which may have an effect. I definitely understand the complexity in this experiment and appreciate the additional experiment done to partially address this issue.

We believe that Figure 4 completely addresses any potential concerns in Figure 3. All the testable hypotheses for how movement differences in Figure 3 could change the main effect were tested and the result was null. These tests are shown in supplemental figure 5.

I think the authors should add a clear 1-2 sentences in the relevant results section where they directly state that there are other parameters such as locomotion or body movements that could explain these results.

We now discuss in the Results that locomotion could have an effect. The reviewer assumes that increased movement (in figure 3) causes an increase in transient response magnitude. Yet, this is unlikely to be true, because tactile perception is known to be suppressed during body movement. Moreover, cortical and thalamic responses to afferent nerve stimulation are weaker during whisking than during rest. Therefore, our main effect in figure 3 is potentially weakened (not enhanced) by greater movement.

See: <https://www.frontiersin.org/articles/10.3389/fpsyg.2013.00913/full>

<https://link.springer.com/article/10.3758/s13423-016-1203-6>

<https://www.jneurosci.org/content/19/17/7603.long>

The authors should also quantify locomotion during self and external touch and present it clearly in the main figure 3 (similar to Figure 3C, D and statistics). In addition, some discussion comparing fixational vision and active whisking would be nice.

We've added the locomotion data to the main figure and discussed vision and active whisking analogy in the revised manuscript.

Typo: Line 40: "... during sensor fixation". Should be sensory

corrected

We thank the reviewer for the careful review, which helped us in improving the quality and readability of our manuscript.

Reviewer #3

The authors addressed my concerns. I wish they did more on the comparison of the current results to responses in other brain regions, but I take their point that this is beyond the scope of the current study. Overall, this is an important paper, because data on somatosensory responses in the SC of awake rodents are missing.

We would like to thank the reviewer for their suggestions and positive comments.